# Neto auxiliary proteins control both the trafficking and biophysical properties of the kainate receptor GluK1

Nengyin Sheng[1], Yun S Shi[1,2], Richa Madan Lomash[3], Katherine W Roche[3], Roger A Nicoll[1,4]*

[1]Deparment of Cellular and Molecular Pharmacology, University of California, San Francisco, San Francisco, United States; [2]The Model Animal Research Center, Key Laboratory of Model Animal for Disease Study of Ministry of Education, Nanjing University, Nanjing, China; [3]Receptor Biology Section, National Institute of Neurological Disorders and Stroke, National Institutes of Health, Bethesda, United States; [4]Department of Physiology, University of California, San Francisco, San Francisco, United States

*For correspondence: roger.nicoll@ucsf.edu

Competing interests: The authors declare that no competing interests exist.

**Abstract** Kainate receptors (KARs) are a subfamily of glutamate receptors mediating excitatory synaptic transmission and Neto proteins are recently identified auxiliary subunits for KARs. However, the roles of Neto proteins in the synaptic trafficking of KAR GluK1 are poorly understood. Here, using the hippocampal CA1 pyramidal neuron as a null background system we find that surface expression of GluK1 receptor itself is very limited and is not targeted to excitatory synapses. Both Neto1 and Neto2 profoundly increase GluK1 surface expression and also drive GluK1 to synapses. However, the regulation GluK1 synaptic targeting by Neto proteins is independent of their role in promoting surface trafficking. Interestingly, GluK1 is excluded from synapses expressing AMPA receptors and is selectively incorporated into silent synapses. Neto2, but not Neto1, slows GluK1 deactivation, whereas Neto1 speeds GluK1 desensitization and Neto2 slows desensitization. These results establish critical roles for Neto auxiliary subunits controlling KARs properties and synaptic incorporation.

## Introduction

Most excitatory synaptic transmission in the brain is mediated by glutamate acting on AMPA and NMDA subtypes of glutamate receptors. However, there is a third subtype of ionotropic glutamate receptor termed kainate receptor (KAR) comprising GluK1-5 subunits. These receptors are unusual in that they are expressed at only a subset of glutamatergic synapses (*Contractor et al., 2011*; *Jane et al., 2009*; *Lerma and Marques, 2013*). The most studied synaptic KARs are those expressed at hippocampal CA3 mossy fiber synapses (*Nicoll and Schmitz, 2005*). These receptors are expressed postsynaptically and generate a slow EPSC. They are also expressed presynaptically and contribute to the profound frequency facilitation, a hallmark of these synapses. In the CA1 region of the hippocampus, KARs are expressed postsynaptically at excitatory synapses in interneurons (*Cossart et al., 1998*; *Frerking et al., 1998*). However, no detectable synaptic KAR EPSCs have been recorded from CA1 pyramidal neurons (*Bureau et al., 1999*; *Castillo et al., 1997*; *Granger et al., 2013*), despite the fact that functional KARs are expressed on these neurons (*Bureau et al., 1999*; *Ruano et al., 1995*). What might determine whether an excitatory synapse expresses KARs?

**eLife digest** Information is transmitted in the brain by cells called neurons. To communicate with neighboring cells, neurons release chemicals called neurotransmitters across a structure called a synapse that forms a junction between the cells. The neurotransmitters bind to receptors on the surface of the receiving neuron, and depending on the type of neurotransmitter released, make that neuron either more or less likely to signal to its neighbors.

Excitatory neurotransmitters make neurons more likely to signal, and glutamate is the most common excitatory neurotransmitter in the brain. There are several different types of receptor that can bind to glutamate, one of which – the kainate receptor – is found at relatively few synapses. These synapses include some in the hippocampus, a region of the brain that is important for memory. Researchers have recently identified two auxiliary proteins, called Neto1 and Neto2, that interact with kainate receptors and appear to affect how strongly the kainate receptors respond when glutamate binds to them. However, the effect of the Neto proteins on one particular subunit of the kainate receptors – called GluK1 – had not been investigated in depth.

CA1 pyramidal neurons are a group of neurons in the hippocampus that are able to produce kainate receptors, but these receptors are not found in CA1 pyramidal neuron synapses. Sheng et al. have now studied CA1 pyramidal neurons from rats, and found that these cells produce a limited amount of GluK1 on their surfaces. However, when GluK1 is expressed together with Neto1 or Neto2, GluK1 receptors appear on the cell surface. Through an independent mechanism Neto proteins also promote the targeting of surface GluK1 to the synapse. Unexpectedly, GluK1 was excluded from synapses that contain another type of glutamate receptor called AMPA receptors.

By measuring the effect of Neto1 and Neto2 on the behavior of GluK1, Sheng et al. found that these proteins modified how the receptor responded to prolonged exposure to glutamate. Specifically, Neto1 increased how quickly GluK1 became desensitized to glutamate, while Neto2 decreased the rate of desensitization. This study demonstrates that Neto proteins play critical roles in controlling the location and biophysical properties of kainate receptors. It will be of interest to see how the present findings apply to other excitatory synapses in the brain.

Recently, auxiliary subunits of KARs, referred to as Neto1 and Neto2, have been identified (*Copits and Swanson, 2012*; *Straub and Tomita, 2012*; *Zhang et al., 2009*). These neurophilin toll-oid-like proteins are single pass transmembrane CUB (complement C1r/C1s, Uegf and Bmp1) domain-containing proteins. Both Neto1 and Neto2 are known to alter the kinetics of KARs (*Copits et al., 2011*; *Straub et al., 2011*; *Zhang et al., 2009*). More specifically Neto2 slows deactivation and desensitization of GluK2 receptors (*Zhang et al., 2009*). Neto1 slows deactivation and desensitization of GluK2/5 and deletion of Neto1 in mice speeds the decay of the KAR-mediated hippocampal mossy fiber EPSC (*Straub et al., 2011*; *Tang et al., 2011*; *Wyeth et al., 2014*). Thus Neto1 can largely explain the biophysical mismatch between heterologously expressed KARs and endogenously expressed KARs. However, the study of the interaction between GluK1 receptor and Neto proteins is limited. It has been reported that Neto1 speeds GluK1 desensitization, whereas Neto2 slows it (*Copits et al., 2011*), but deactivation was not examined. Although the primary role of Neto proteins appears to be the modulation of KAR function, their role in receptor trafficking is less clear. Neto2 has no effect on the surface expression of GluK2 in oocytes (*Zhang et al., 2009*), although it has been reported to enhance surface expression of GluK1 in HEK cells and cultured neurons (*Copits et al., 2011*). The knock-out of Neto1 in mice does not alter the neuronal surface expression or synaptic localization of GluK2/5 (*Straub et al., 2011*), although other studies reported a decrease in PSD expression of GluK2 when Neto1 was knocked-out (*Tang et al., 2011*; *Wyeth et al., 2014*). Finally, it has been reported that Neto2 can target GluK1 to synapses of primary cultured neurons (*Copits et al., 2011*; *Palacios-Filardo et al., 2014*). However, it remains controversial whether Neto1 and Neto2 are required for the surface and synaptic expression of GluK1 receptor. If so, it remains unclear whether the bases for these two trafficking steps are the same or not.

The lack of endogenous expression of GluK1 in CA1 pyramidal neuron provides a null background in which to study the rules governing GluK1 function. Indeed, recent studies have shown that expression of GluK1 and Neto2 results in the appearance of KAR synaptic currents (*Copits et al., 2011*; *Granger et al., 2013*; *Palacios-Filardo et al., 2014*). Therefore, we have selected the CA1 neuron as a model to study the roles of Neto1 and Neto2 in the surface and synaptic trafficking and kinetics of the GluK1 receptor.

## Results

### Synaptic trafficking of GluK1 receptors is dependent on Neto1 and Neto2 proteins

CA1 pyramidal neurons express functional kainate receptors (*Bureau et al., 1999*). However, no detectable synaptic KAR-mediated responses can be detected (*Bureau et al., 1999*; *Castillo et al., 1997*; *Granger et al., 2013*). We wondered if the lack of synaptic responses might be due to a limited expression of the auxiliary subunit Neto1 or Neto2 (*Ng et al., 2009*; *Palacios-Filardo et al., 2014*). We therefore expressed these proteins exogenously in CA1 neurons of cultured rat hippocampal slices through biolistic transfection and measured the synaptic responses by dual whole-cell recordings. Neither Neto1 (*Figure 1A1*) nor Neto2 (*Figure 1B1*) had any effect on the size of the synaptic response recorded at −70 mV or the NMDA receptor (NMDAR) response recorded at +40 mV (*Figure 1A2 and B2*). Moreover, overexpression had no effect on paired-pulse ratio, a measure of presynaptic release probability (Neto1 vs control: 1.29 ± 0.11 vs 1.23 ± 0.08, p>0.05; Neto2 vs control: 1.56 ± 0.08 vs 1.46 ± 0.1, p>0.05). One possibility is that KARs were recruited to the synapse, but that they replaced synaptic AMPA receptors (AMPARs). To test this possibility we applied the AMPAR selective antagonist GYKI53655. The antagonist completely blocked the responses both in Neto1 (*Figure 1A1*) and Neto2 (*Figure 1B1*) expressing neurons, suggesting that Neto proteins cannot promote incorporation of the endogenous KARs into synapses. It should be noted that in dissociated neuronal cultures expression of Neto1 or Neto2 generated infrequent KAR mediated synaptic responses in a small fraction of cells (*Palacios-Filardo et al., 2014*).

Perhaps the lack of synaptic KARs is due to the limited expression of these receptors in these neurons. We thus expressed GluK1, but this did not affect the size of the response recorded at −70 mV (*Figure 1C1*) or the NMDAR response (*Figure 1C2*), as well as the paired-pulse ratio (GluK1 vs control: 1.89 ± 0.17 vs 1.89 ± 0.22, p>0.05). Furthermore, GYKI53655 fully blocked the EPSCs indicating that functional KARs were not recruited to the synapse. We next expressed GluK1 together with Neto1 and in this case there was a large increase in the size of synaptic response recorded at −70 mV and GYKI53655 only partially blocked the response (*Figure 2A*). We selected a concentration of 100 μM GYKI53655 to ensure that all AMPARs were blocked (*Bleakman et al., 1996*). This concentration, however, will block approximately 20% of KAR mediated responses (*Bleakman et al., 1996*) and thus the currents remaining in GYKI53655 underestimate the actual contribution of GluK1 receptors to synaptic transmission. These experiments were repeated by expressing GluK1 along with Neto2. As reported previously (*Granger et al., 2013*) the synaptic response was greatly increased and GYKI53655 only partially blocked the response (*Figure 2B*). Although presynaptic KARs are known to regulate glutamate release at mossy fibers, sparse expression of GluK1 receptors in CA1 neurons has no effect on presynaptic release probability as there is no significant change of paired-pulse ratio (GluK1/Neto1 vs control: 1.46 ± 0.18 vs 1.7 ± 0.26, p>0.05; GluK1/Neto2 vs control: 1.36 ± 0.12 vs 1.52 ± 0.16, p>0.05). These findings and those in *Figure 1* are summarized in *Figure 2C*, showing that synaptic KAR responses are only observed when GluK1 is expressed along with either Neto1 or Neto2. To determine if the synaptic delivery of KARs is depended on synaptic activity, we incubated the cultured slices in NBQX and AP5 to inhibit AMPARs and NMDARs activation during the expression of KARs. We then compared the evoked synaptic responses between experimental and control neurons. However, the receptor antagonists did not prevent the synaptic incorporation of either GluK1/Neto1 or GluK1/Neto2 (*Figure 2—figure supplement 1*), suggesting that synaptic activity is not required for Neto-dependent GluK1 synaptic trafficking.

Interestingly, when GluK1 was expressed along with Neto1 or Neto2 there was a significant increase in the size of the NMDAR EPSCs (*Figure 2—figure supplement 2A–C*). We further confirm the increased NMDAR-mediated synaptic response with recordings done in the presence of NBQX

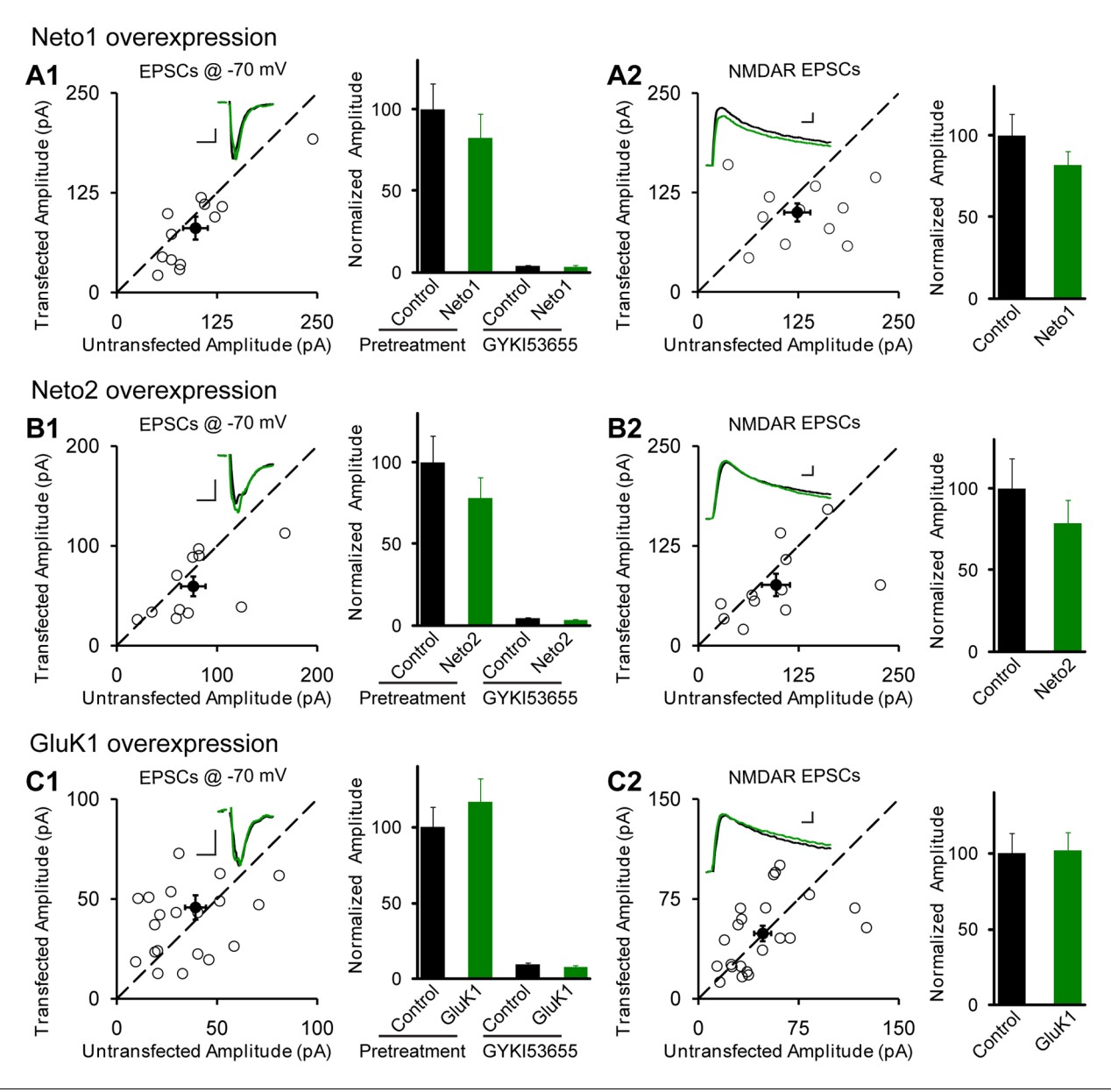

**Figure 1.** Individual overexpression of Neto1, Neto2 or GluK1 has no effect on synaptic transmission. Rat hippocampal slice cultures were biolistically transfected with Neto1 (A, n=12), Neto2 (B, n=11) or GluK1 (C, n=22). Simultaneous dual whole-cell recordings from a transfected CA1 pyramidal neuron (green trace) and a neighboring wild type one (black trace) were performed. The evoked EPSCs (eEPSCs) were measured at −70 mV and +40 mV (the current amplitudes were measured 100 ms after stimulation). Open and filled circles represent amplitudes for single pairs and mean ± SEM, respectively. Insets show sample current traces from control (black) and experimental (green) cells. The scale bars for representative eEPSC trace were 25 pA and 25 ms. Bar graphs show normalized eEPSC amplitudes (mean ± SEM) of −70 mV (A1, 82.24 ± 14.64% control, p > 0.05; B1, 77.9 ± 12.9% control, p > 0.05 and C1, 116.58 ± 15.53% control, p > 0.05) and +40 mV (A2, 81.74 ± 8.42% control, p > 0.05; B2, 78.53 ± 14.35% control, p > 0.05 and C2, 101.8 ± 12.06% control, p > 0.05) presented in scatter plots. All the statistical analyses are compared to respective control neurons with two-tailed Wilcoxon signed-rank sum test. The eEPSC amplitudes measured at −70 mV after GYKI53655 (100 µM) wash-in in A1, B1 and C1 were also normalized according to respective pretreated control neurons.

to block AMPAR and KAR EPSCs (*Figure 2—figure supplement 2D*). These results raise the possibility that GluK1 together with Netos has a synaptogenic effect. We, therefore, filled neurons with Alexa Fluor 568 dye and analyzed the density of dendritic spines as a proxy for the density of

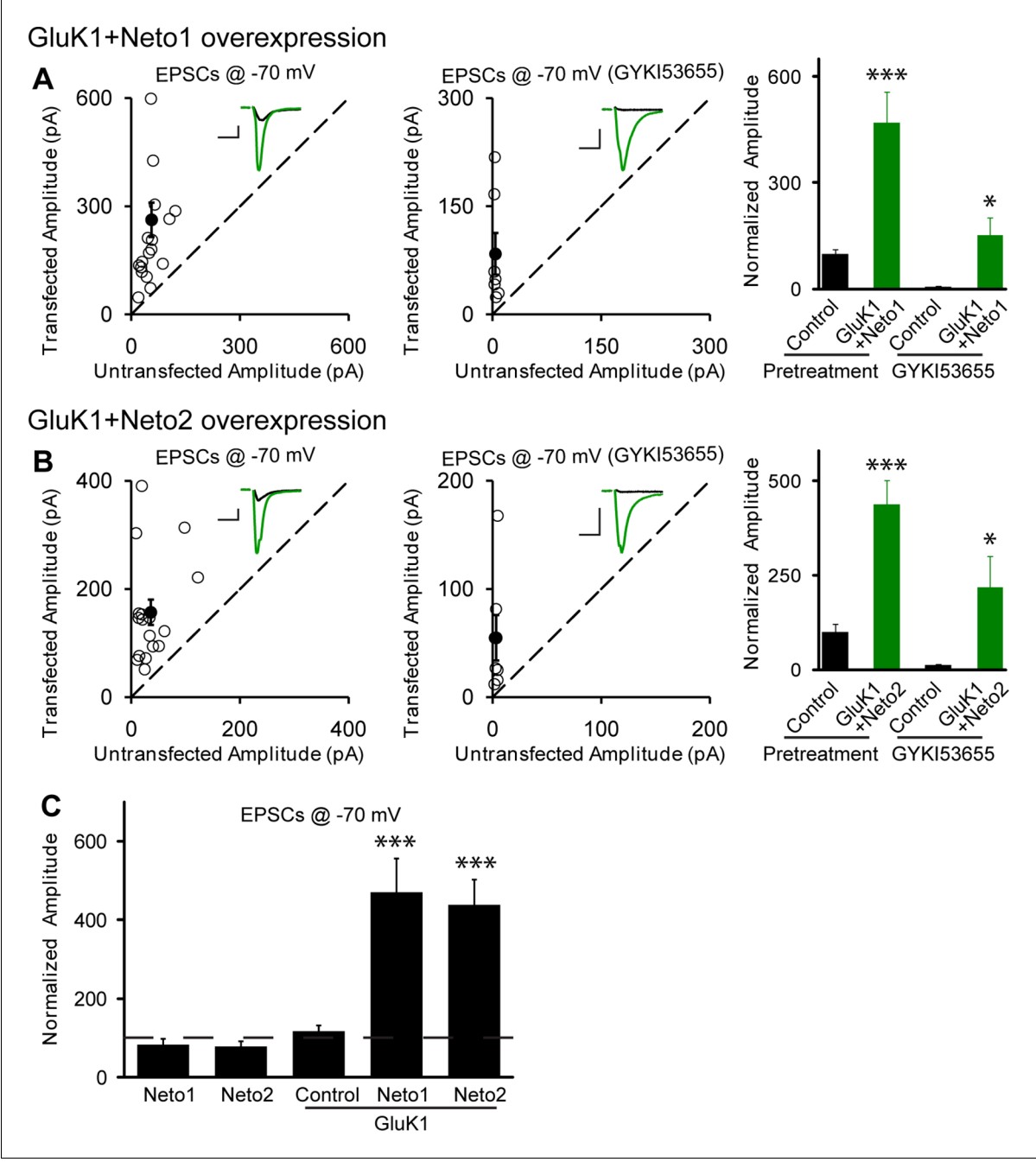

**Figure 2.** Neto1 and Neto2 promote GluK1 receptor synaptic targeting. (**A**) Scatter plots show eEPSC amplitudes of control and GluK1/Neto1-cotransfected CA1 neurons in rat hippocampal slice cultures measured at −70 mV in the absence or presence of GYKI53655. Filled circles show mean ± SEM. Insets show sample current traces from control (black) and GluK1/Neto1-expressing (green) cells. The scale bars for representative eEPSC trace were 50 pA and 25 ms. Bar graph show normalized eEPSC amplitudes (mean ± SEM) of pretreated (n=19, 470.65 ± 85.6% control, *** p < 0.0005) and GYKI53655 treated (n=7, 150.72 ± 51.8% control pretreatment, * p < 0.05) cells. (**B**) Scatter plots show eEPSC amplitudes of control and GluK1/Neto2-cotransfected CA1 neurons in rat hippocampal slice cultures measured at −70 mV in the absence or presence of GYKI53655. Filled circles show mean ± SEM. Insets show sample current traces from control (black) and GluK1/Neto1-expressing (green) cells. The scale bars for representative eEPSC trace were 50 pA and 25 ms. Bar graph show normalized eEPSC amplitudes (mean ± SEM) of pretreated (n=17, 689.52 ± 195.16% control, *** p < 0.0005) and GYKI53655 treated (n=7, 317.63 ± 83.12% control pretreatment, * p < 0.05) cells. (**C**) Summary of the normalized evoked EPSC amplitudes at −70 mV as percent of respective control ± SEM for each indicated transfection. All the statistical analyses are compared to respective control neurons with two-tailed Wilcoxon signed-rank sum test.

The following figure supplements are available for figure 2:

*Figure 2 continued on next page*

*Figure 2 continued*

**Figure supplement 1.** Neto1 and Neto2 regulation of GluK1 synaptic expression is independent of synaptic activity.

**Figure supplement 2.** NMDAR EPSCs are increased in GluK1/Neto1 and GluK1/Neto2 expressing neurons.

excitatory synapses (*Figure 3A and B*). However, we found no difference in spine density in neurons expressing GluK1/Neto1 or GluK1/Neto2 compared to control.

## Neto proteins specifically target GluK1 receptors to silent synapses

How are KARs incorporated into synapses? Do they displace synaptic AMPARs or do they add additional receptors to the already activated synapse? To address these questions we expressed GluK1 with either Neto1 (*Figure 3C*) or Neto2 (*Figure 3D*) and recorded synaptic responses in the presence of the GluK1 selective antagonist ACET. In this case there was no significant difference in the AMPAR-mediated responses between control and experimental neurons, indicating that AMPARs are not displaced by the synaptic expression of KARs.

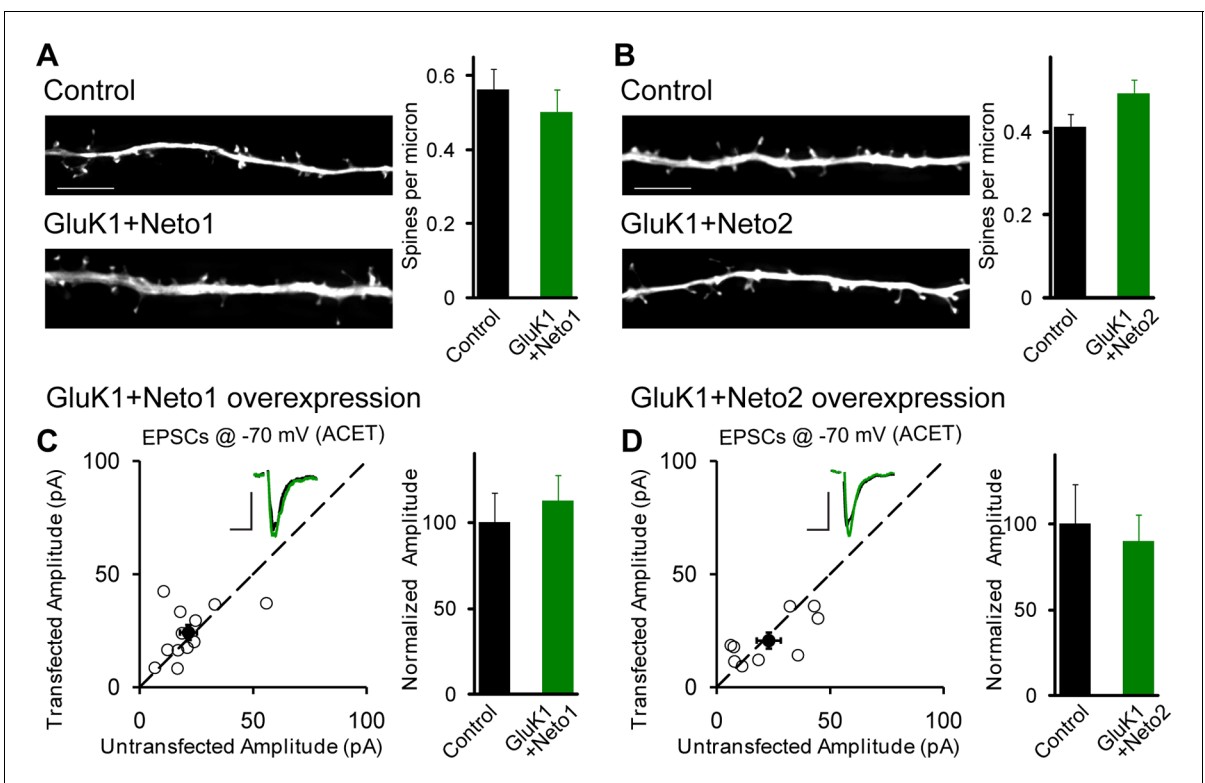

**Figure 3.** GluK1 synaptic expression has no effect on spinogenesis and does not replace endogenous synaptic AMPA receptors. (**A**) Sample images of primary apical dendrites from control (upper) and GluK1/Neto1 overexpressed (lower) neurons imaged using super-resolution structured illumination microscopy (SIM). Bar graph in right shows average spine density (control, n = 8, 0.56 ± 0.06/μm; GluK1/Neto1, n = 9, 0.5 ± 0.06/μm; p > 0.05). Scale bar: 5 μm. (**B**) Sample images of primary apical dendrites from control (upper) and GluK1/Neto2 overexpressed (lower) neurons imaged using SIM. Bar graph in right shows average spine density (control, n = 8, 0.41 ± 0.03/μm; GluK1/Neto2, n = 7, 0.49 ± 0.03/μm; p > 0.05). Scale bar: 5 μm. All the statistical analyses are compared to respective control neurons with unpaired two-tailed t test. (**C and D**) Scatter plots show eEPSC amplitudes of control and GluK1/Neto1 (**C**) or GluK1/Neto2 (**D**) cotransfected neurons in rat hippocampal slice cultures measured at −70 mV in the presence of the GluK1 antagonist ACET (1 μM). Filled circles show mean ± SEM. Insets show sample current traces from control (black) and experimental (green) cells. The scale bars for representative eEPSC trace were 25 pA and 25 ms. Bar graph show normalized eEPSC amplitudes (mean ± SEM) (A: n=12, 112.33 ± 15.36% control, p > 0.05; B: n=9, 89.7 ± 15.38% control, p > 0.05) presented in scatter plots. All the statistical analyses are compared to respective control neurons with two-tailed Wilcoxon signed-rank sum test.

There are two possible explanations for the results in *Figure 2* and *Figure 3*. Either the expressed KARs populate synapses that already express AMPARs (*Figure 4—figure supplement 1A*) or they are excluded from AMPAR expressing synapses and selectively populate silent synapses, i.e. those that do not express AMPARs (*Figure 4—figure supplement 1B*). In the former situation one would expect the size of quantal events to be larger, whereas in the latter case one might expect to see primarily a change in frequency. To test these predictions we replaced $Ca^{2+}$ with $Sr^{2+}$, which desynchronizes the induced transmitter release (*Oliet et al., 1996*). We then simultaneously recorded from a control cell and an experimental one expressing GluK1 and Neto1 (*Figure 4A*) or Neto2 (*Figure 5A*) to examine the amplitude and frequency of asynchronous EPSCs (aEPSCs). In cells expressing GluK1 and Neto1, we observed no change in quantal size (*Figure 4A2* and *Figure 4—figure supplement 2A1*), but a large increase in frequency (*Figure 4A3* and *Figure 4—figure supplement 2A2*). We observed the same results when expressing GluK1 and Neto2 (*Figure 5A1-A3* and *Figure 4—figure supplement 2C1-C2*). These results tell us that expression of KARs results in synapse unsilencing. The finding that the AMPAR EPSC in GluK1 expressing cells is not reduced in ACET indicates that KARs do not cause a net loss of synaptic AMPARs (*Figure 3C and D*). However, there remains a possibility that synaptic KARs are expressed at all synapses, displacing a portion of synaptic AMPARs to previously silent synapses (*Figure 4—figure supplement 1C*). In this scenario each synapse would contain a mixture of AMPARs and KARs, and selectively inhibiting KARs activity would be expected to reduce the size of aEPSCs. To test this idea, we repeated the asynchronous electrophysiological recordings in the presence of the GluK1 antagonist ACET. In the presence of ACET the increase in aEPSC frequency in the GluK1/Neto1-expressing neurons (*Figure 4A3*) was no longer observed (*Figure 4B2-C2* and *Figure 4—figure supplement 2B2*). Importantly we saw no reduction in the size of aEPSCs (*Figure 4C1* and *Figure 4—figure supplement 2B1*). We found the same results with GluK1/Neto2 expressing neurons (*Figure 5B2-B3*, *Figure 4—figure supplement 2D1-2D2*). The lack of change in aEPSC size in ACET (*Figure 4C1*, *Figure 5B2*, *Figure 4—figure supplement 2B1* and *Figure 4—figure supplement 2D1*), indicates that synaptic GluK1 and AMPARs do not co-localize at the same synapses.

We also examined the effect of GYKI53655 on aEPSCs and found that at a relative low concentration (30 µM) the aEPSCs from control cells were totally blocked, while a large reduction in aEPSC frequency and minimal reduction in aEPSC amplitude from GluK1/Neto1-transfected cells (*Figure 4D1-D4*) were observed. Taken together, these results suggest that KARs are excluded from synapses that are already populated with AMPARs. Rather they appear to selectively populate synapses that lack AMPARs, i.e. silent synapses. Perhaps even more intriguing is that the average size of GluK1-mediated quantal events is the same as the AMPAR-mediated events. This implies that the average single channel conductance and number of receptors at GluK1 synapses is the same as that for AMPAR expressing synapses, or more likely, that there is some type of homeostatic process that governs the number of synaptic KARs.

## The structural basis for Neto1 and Neto2 regulation of GluK1 synaptic trafficking

Our above results indicate that synaptic trafficking of GluK1 receptors is dependent on Neto proteins. We next examined which region(s) in Neto proteins are responsible for targeting GluK1 to the synapse. As Neto1 and Neto2 are single transmembrane proteins, we first tested the involvement of their intracellular domains. Deletion of the entire C-tail (Δ161) of Neto1 prevents the targeting of GluK1 receptors to the synapse (*Figure 6A,B and H*) whereas deletion of the last 4 amino acids (Δ4), a putative PDZ binding motif, has no significant effect (*Figure 6F and H*). Deletion of the last 20 amino acids (Δ20) of Neto1 has a substantial effect on synaptic GluK1 currents (*Figure 6E and H*), as did a larger deletion of 41 amino acids (Δ41) (*Figure 6D and H*). These truncation experiments suggest that the last 20 amino acids of Neto1 are critical for the synaptic incorporation of functional GluK1 receptors. It has been reported that the AMPAR auxiliary subunit stargazin can be phosphorylated in the intracellular C-tail, which regulates its interaction with negatively charged lipid bilayers and therefore synaptic AMPAR activity (*Sumioka et al., 2010*; *Tomita et al., 2005*). We therefore mutated three serines and a tyrosine in this region to alanines simultaneously (S3Y/A) and found that the Neto1S3Y/A mutant disrupted GluK1 synaptic expression to the same extent as deleting the last 20 amino acids (*Figure 6G and H*).

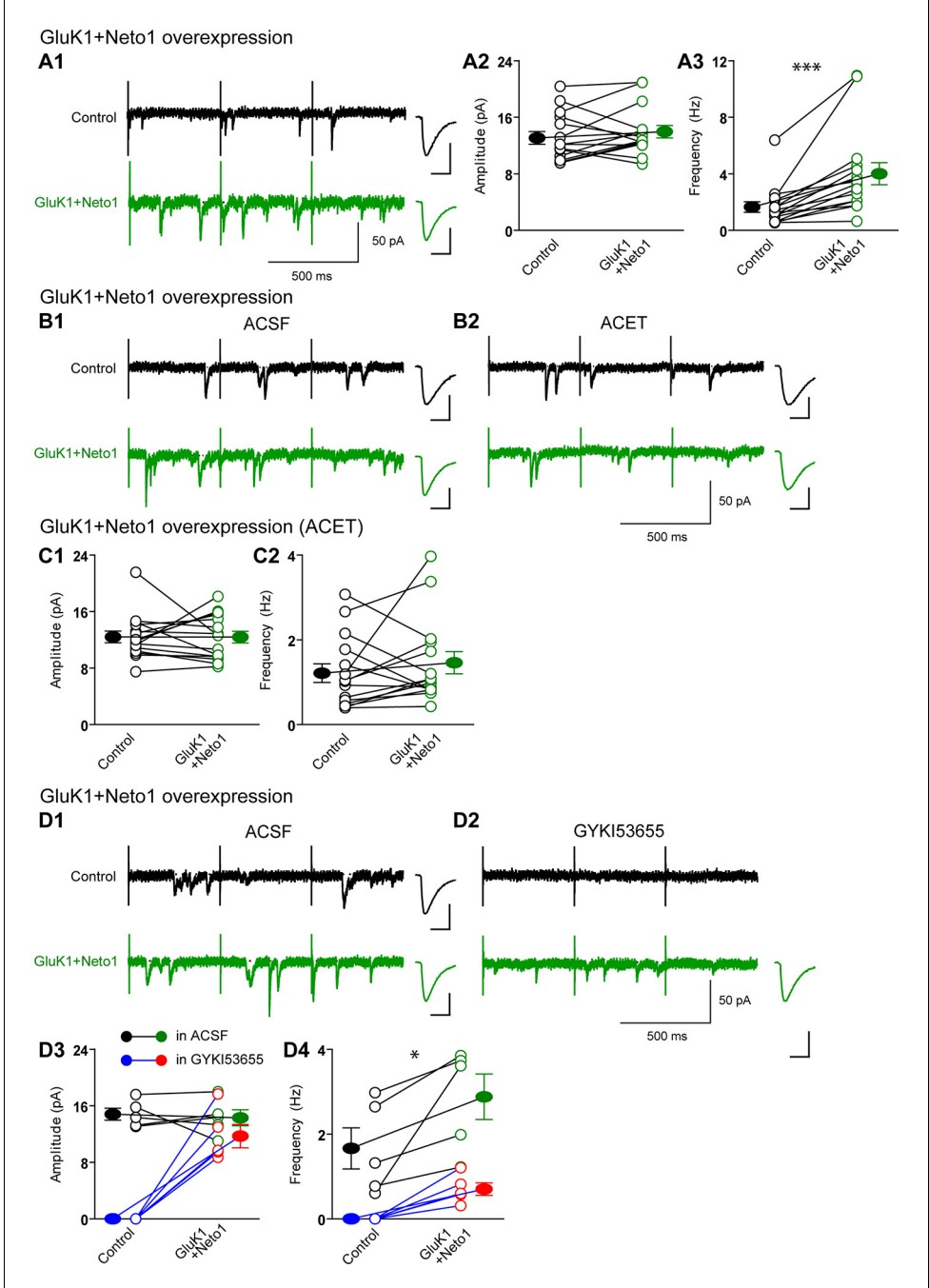

**Figure 4.** Neto1 specifically targets GluK1 receptors to silent synapse. (**A1**) Representative sample traces of asynchronous EPSCs (aEPSCs) simultaneously recorded in the presence of $Sr^{2+}$ from control (black) and GluK1/Neto1-coexpressed (green) neurons. The first 50 ms following stimulation was excluded from analysis. The scale bars for single representative aEPSC traces were 10 pA and 10 ms. (**A2**) aEPSC amplitude is not significantly changed in GluK1/Neto1-expressing neurons (n=15, control: 13.08 ± 0.89 pA, GluK1/Neto1: 13.96 ± 0.89 pA, p > 0.05). Plot shows single pairs (open circles) and mean ± SEM (filled circles). (**A3**) aEPSC frequency is significantly increased in neurons expressing GluK1 and Neto1 (n=15, control: 1.65 ± 0.38 Hz, GluK1/Neto1: 4.01 ± 0.79 Hz, *** p < 0.0001). Plot shows single pairs (open circles) and mean ± SEM (filled circles). (**B1 and B2**) Representative sample traces of aEPSCs recorded in the presence of $Sr^{2+}$ from control (black) and GluK1/Neto1-coexpressed (green) neurons before (**B1**, left) and after (**B2**, right) ACET treatment. The first 50 ms following stimulation was excluded from analysis. The scale bars for single representative aEPSC trace were 10 pA and 10 ms. (**C1**) Plot shows single pairs (open circles) and mean ± SEM (filled circles) of aEPSC amplitude from control and GluK1/Neto1 transfected neurons (n=15, control: 12.41 ± 0.82 pA, GluK1/Neto1: 12.38 ± 0.83 pA, p > 0.05) recorded in the presence of ACET. (**C2**) The aEPSC frequency in neurons expressing GluK1 and Neto1 is not significantly different from control ones in the presence of ACET (n=15, control: 1.21 ± 0.22 Hz, GluK1/Neto1: 1.46 ± 0.26 Hz, p > 0.05). Plot shows single pairs (open circles) and mean ± SEM (filled circles). (**D1 and D2**) Representative

*Figure 4 continued on next page*

*Figure 4 continued*

sample traces of aEPSCs recorded in the presence of Sr$^2$ from control (black) and GluK1/Neto1-coexpressed (green) neurons before (**D1**, left) and after (**D2**, right) GYKI53655 (30 µM) treatment. The first 50 ms following stimulation was excluded from analysis. The scale bars for single representative aEPSC trace were 10 pA and 10 ms. (**D3-D4**) Plots show single paired (open circles) and mean ± SEM (filled circles) of aEPSC amplitude (**D3**) and frequency (**D4**) from control and GluK1/Neto1-cotransfected neurons before (black and green, n=5; amplitude: control: 14.81 ± 0.85 pA, GluK1/Neto1: 14.3 ± 1.13 pA, p > 0.05; frequency: control: 1.66 ± 0.19 Hz, GluK1/Neto1: 2.88 ± 0.54 Hz, p < 0.05) and after 30 µM GYKI53655 treatment (blue and red, amplitude: GluK1/Neto1: 11.71 ± 1.66 pA; frequency: GluK1/Neto1: 0.7 ± 0.15 Hz). All the statistical analyses are compared to respective control neurons with two-tailed Wilcoxon signed-rank sum test.

The following figure supplements are available for figure 4:

**Figure supplement 1.** The models of Neto proteins-regulated synaptic GluK1 receptors localization.

**Figure supplement 2.** GluK1 receptors specifically traffic to silent synapses in the presence of Neto1 or Neto2.

We next looked for the domain(s) in Neto2 that are critical for GluK1 synaptic trafficking. Deletion of the entire C-terminal domain (Δ148) fully diminished GluK1 receptor synaptic targeting by Neto2 (*Figure 7A,B and I*) whereas deleting the last 4 amino acids (Δ4) has no effect on the function of

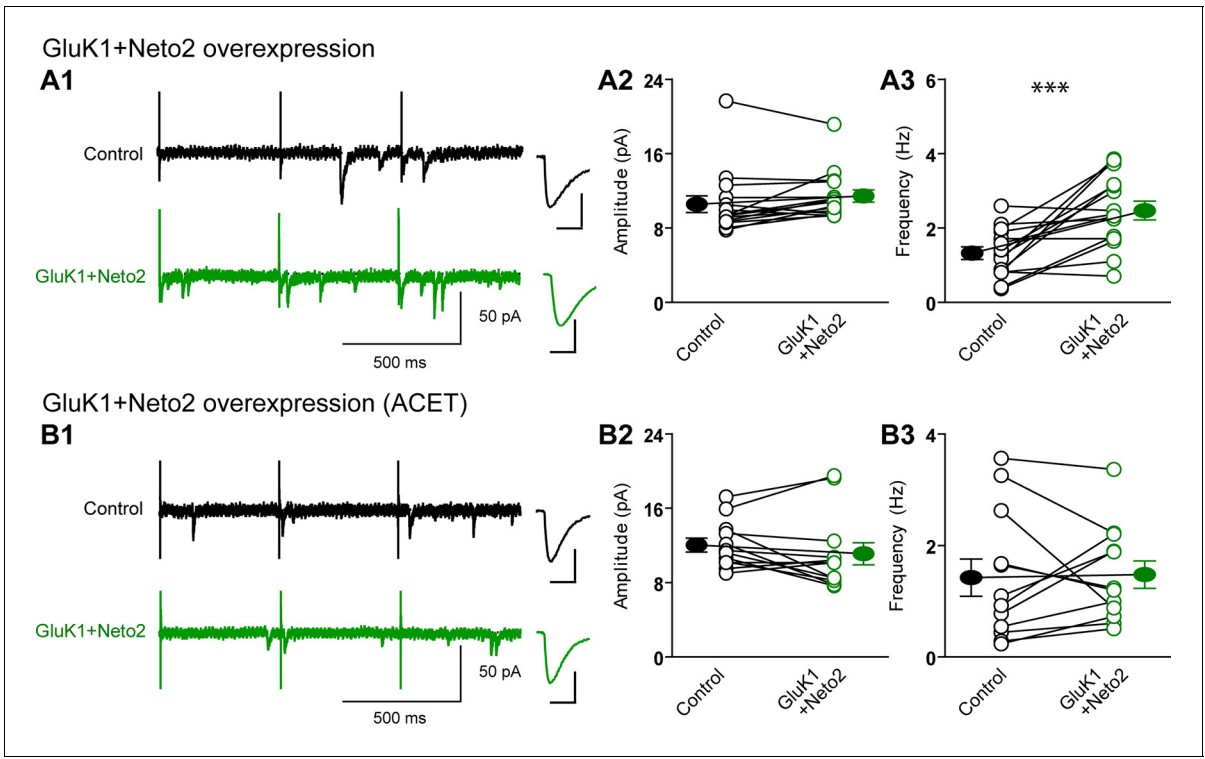

**Figure 5.** Neto2 specifically targets GluK1 receptors to silent synapse. (**A1**) Representative sample traces of aEPSCsfrom control (black) and GluK1/Neto2-coexpressed (green) neurons. The first 50 ms following stimulation was excluded from analysis. The scale bars for single representative aEPSC trace were 10 pA and 10 ms. (**A2**) aEPSC amplitude is not significantly changed in GluK1/Neto2 expressed neurons (n=15, control: 10.56 ± 0.89 pA, GluK1/Neto2: 11.44 ± 0.67 pA, p > 0.05). Plot shows single pairs (open circles) and mean ± SEM (filled circles). (**A3**) aEPSC frequency is significantly increased in neurons expressing GluK1 and Neto2 (n=15, control: 1.32 ± 0.17 Hz, GluK1/Neto2: 2.45 ± 0.26 Hz, *** p < 0.0005). Plot shows single pairs (open circles) and mean ± SEM (filled circles). (**B1**) Representative sample traces of aEPSCs simultaneously recorded in the presence of Sr$^{2+}$ and ACET from control (black) and GluK1/Neto2-coexpressed (green) neurons. The first 50 ms following stimulation was excluded from analysis. The scale bars for single representative aEPSC trace were 10 pA and 10 ms. (**B2**) Plot shows single pairs (open circles) and mean ± SEM (filled circles) of aEPSC amplitude from control and GluK1/Neto2 transfected neurons (n=12, control: 12.03 ± 0.74 pA, GluK1/Neto2: 11.1 ± 1.19 pA, p > 0.05). (**B3**) The aEPSC frequency in neurons expressing GluK1 and Neto2 is not significantly different from control ones in the presence of ACET (n=12, control: 1.42 ± 0.33 Hz, GluK1/Neto1: 1.48 ± 0.25 Hz, p > 0.05). Plot shows single pairs (open circles) and mean ± SEM (filled circles). All the statistical analyses are compared to respective control neurons with two-tailed Wilcoxon signed-rank sum test.

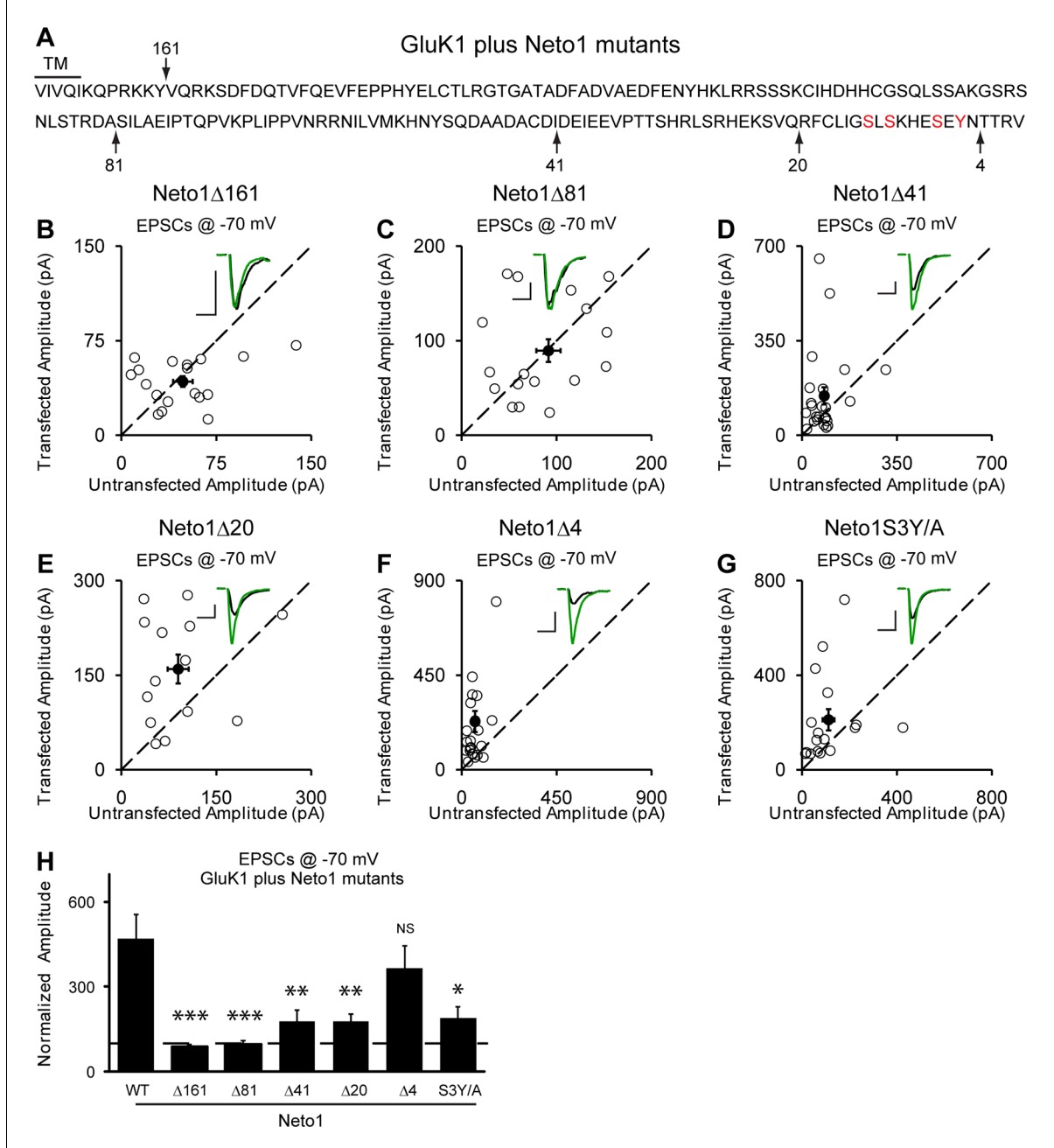

**Figure 6.** Neto1-mediated GluK1 synaptic trafficking is dependent on the critical serine and tyrosine residues in the intracellular region. (**A**) Amino acid sequence of the Neto1 C-tail. The truncation mutants generated are indicated by arrows. The Neto1S3Y/A is a mutant in which the highlighted three serines and one tyrosine residues within the last 20 amino acids are mutated to alanines. (**B–G**) Scatter plots of eEPSCs at −70 mV for GluK1 co-expressed with various Neto1 mutants. Open circles are individual pairs and filled are mean ± SEM. Insets show sample current traces from control (black) and experimental (green) cells. The scale bars for representative eEPSC trace were 50 pA and 25 ms. (**H**) Normalized evoked EPSC amplitudes at −70 mV as percent of respective control ± SEM for each transfection (GluK1/Neto1: n=19, 470.65 ± 85.59% control; GluK1/Neto1Δ161: n=18, 87.7 ± 8.67% control, *** p < 0.0001; GluK1/Neto1Δ81: n=18, 97.8 ± 13.1% control, *** p < 0.0001; GluK1/Neto1Δ41: n=24, 179.2 ± 39.0% control, ** p < 0.005; GluK1/Neto1Δ20: n=14, 178.1 ± 25.5% control, ** p < 0.005; GluK1/Neto1Δ4: n=23, 365.62 ± 80.22% control, p > 0.05; GluK1/Neto1S3Y/A: n=17, 189.48 ± 40.26% control, * p < 0.05). All the statistical analyses are tested with the group co-overexpressing GluK1 and wildtype Neto1 using Mann-Whitney U-test.

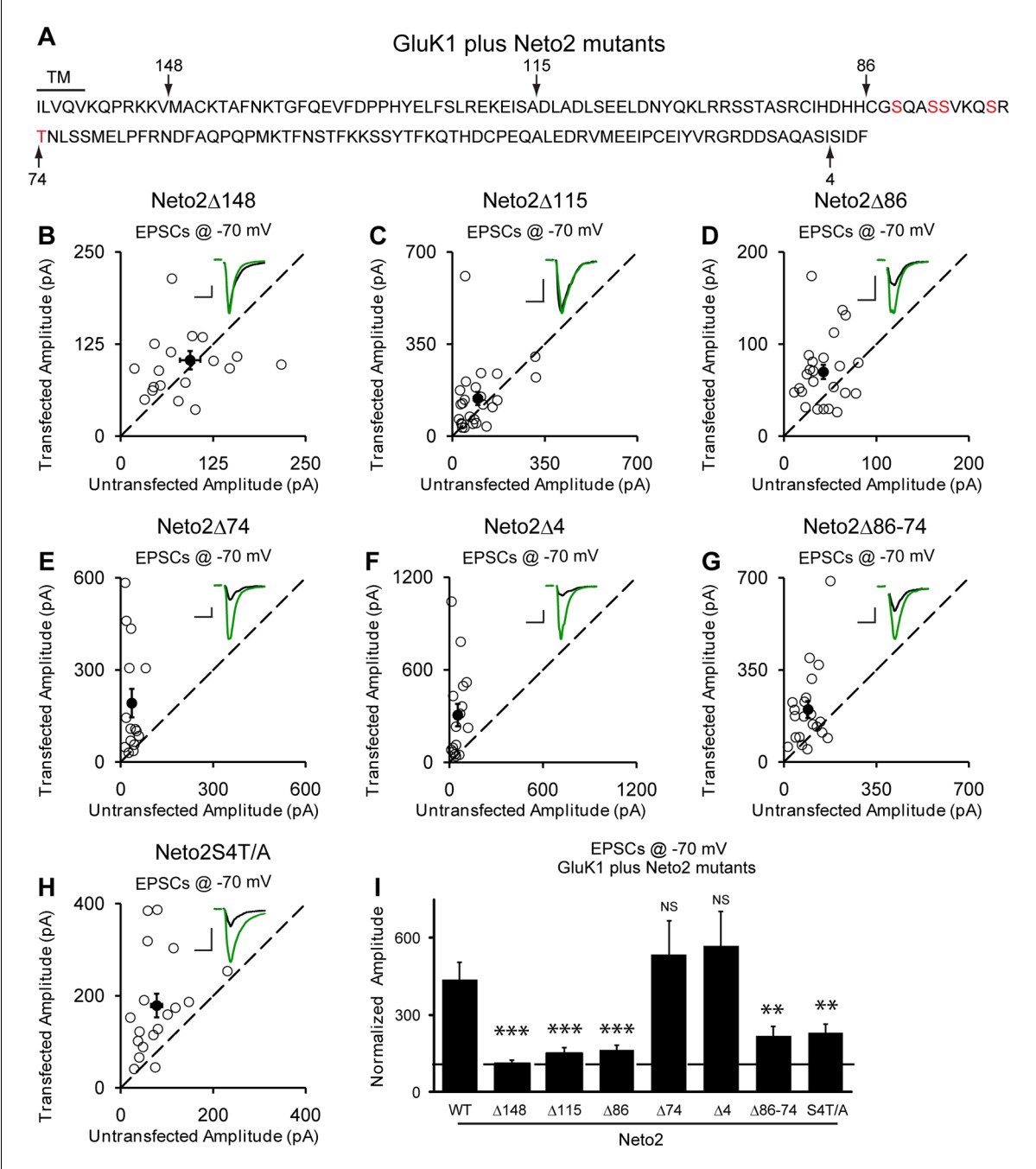

**Figure 7.** Neto2-mediated GluK1 synaptic trafficking is dependent on the critical serine and threonine residues in the intracellular region. (**A**) Amino acid sequence of the Neto2 C-tail. The truncation mutants generated are indicated by arrows. The Neto2S4T/A is a mutant in which the highlighted four serines within region 86-74 and one threonine just next to this region are mutated to alanines. (**B–H**) Scatter plots of eEPSCs at −70 mV for GluK1 co-expressed with various Neto2 mutants. Open circles are individual pairs and filled are mean ± SEM. Insets show sample current traces from control (black) and experimental (green) cells. The scale bars for representative eEPSC trace were 50 pA and 25 ms. (**I**) Normalized evoked EPSC amplitudes at −70 mV as percent of respective control ± SEM for each transfection (GluK1/Neto2: n=17, 473.69 ± 65.08% control; GluK1/Neto2Δ148: n=19, 109.96 ± 13.27% control, *** p < 0.0001; GluK1/Neto2Δ115: n=25, 148.51 ± 25.59% control, *** p < 0.0005; GluK1/Neto2Δ86: n=24, 163.72 ± 18.08% control, *** p < 0.0005; GluK1/Neto2Δ74: n=15, 535.85 ± 129.35% control, p > 0.05; GluK1/Neto2Δ4: n=16, 567.38 ± 134.55% control, p > 0.05; GluK1/Neto2Δ86–74: n=21, 219.09 ± 35.39% control, ** p < 0.01; GluK1/Neto2S4T/A: n=18, 229.98 ± 32.99% control, ** p < 0.01). All the statistical analyses are tested with the group co-overexpressing GluK1 and wildtype Neto1 using Mann-Whitney U-test.

Neto2 (*Figure 7F and I*). Moreover, deletions of 115 amino acids (Δ115) (*Figure 7C and I*) and 86 amino acids (*Figure 7D and I*) also significantly impaired the function of Neto2. These results indicate that the last 86 amino acids of Neto2 are critical for the incorporation of synaptic GluK1 receptors. To narrow down the critical region of Neto2, we carried out further deletions within the last 86 amino acids. Of particular importance is the region between 86 and 74 amino acids (Δ86-74) as its deletion significantly impaired Neto2-regulated GluK1 synaptic targeting (*Figure 7G and I*), whereas deletion of last 74 amino acids (Δ74) has no significant effect on GluK1-mediated synaptic response (*Figure 7E and I*). Furthermore, mutation of the serines in this region as well as the threonine just next to this region to alanines (S4T/A) also impaired GluK1 synaptic expression to the same extent as this critical deletion mutant (Δ86-74) (*Figure 7H and I*).

## Regulation of GluK1 surface expression and biophysical properties by Neto proteins

The synaptic delivery of KARs involves at least two steps. The receptors first have to be properly assembled and delivered to the surface, followed by targeting of the surface receptors to the synapse. As it has been reported that Neto2 increases GluK1 surface expression (*Copits et al., 2011*), Neto proteins may simply increase the pool of surface receptors to such an extent that the receptors passively populate synapses. To test this possibility, we measured surface GluK1 expression electrophysiologically by pulling outside-out membrane patches from the soma, with the goal of determining whether Neto mutants that had impaired KAR synaptic localization were simply unable to increase surface KAR trafficking. Since KAR currents desensitized rapidly, glutamate was applied using ultra-fast perfusion. All currents were recorded in the presence of GYKI53655 (100 μM) to block AMPAR-mediated response. In wild type patches from CA1 neurons, we were unable to detect any glutamate-evoked current (*Figure 8A*). This contrasts to KAR-mediated currents recorded in a whole cell recording configuration with bath application of agonist (*Bureau et al., 1999*). Presumably the low density of these receptors accounts for the lack of current in outside-out patches. In neurons expressing only GluK1 we saw small, but significant, glutamate evoked currents whereas in patches from neurons co-expressing either Neto1 or Neto2 with GluK1 very large currents were recorded (*Figure 8A*). We then examined the Neto mutants that greatly impaired synaptic responses and looked for the Neto mutants to modulate KAR surface expression. If there were an additional targeting role for Netos we would expect some of these mutants, which impaired GluK1-mediated synaptic responses, to generate extrasynaptic KAR currents similar in magnitude to that recorded when wild type Netos were expressed with GluK1. Indeed, both Neto1S3Y/A and Neto2S4T/A mutants generated currents in outside-out patches of similar size to those generated by wild type Neto1 and Neto2 (*Figure 8A*). These results provide strong evidence that there is, in fact, a role for these auxiliary proteins in targeting surface GluK1 receptors to synapses.

It has been shown that Neto2 has no effect on the surface delivery of the GluK2 receptors but instead causes a large increase in GluK2-evoked current by changing the gating properties of the receptors (*Zhang et al., 2009*). Thus it was important to determine if Netos actually increase the surface expression of GluK1. We therefore transfected dissociated hippocampal neurons with an HA-tagged GluK1 receptor alone or together with Netos and examined surface expression of GluK1. In the absence of Netos, the surface labeling was very weak, although the neuron clearly expressed GluK1 (*Figure 8B*). However, in the presence of either Neto1 or Neto2, GluK1 was abundantly expressed on the surface (*Figure 8B and C*). Consistently, GluK1-specific current could be observed through whole-cell puffing with glutamate or kainic acid in the present of Neto1 auxiliary subunit (*Figure 8—figure supplement 1A*). Together, these results indicate that both Neto1 and Neto2 can drive the robust surface expression of GluK1. However, the tagged-GluK1 receptors cannot traffic to synapse as the synaptic responses were not increased even coexpressing HA-GluK1 or Myc-GluK1 together with Neto1 protein (*Figure 8—figure supplement 1B*). To test whether the GluK1 receptor indeed localized at the synapse, we used a tagged-Neto2 mutant which promote GluK1 synaptic expression efficiently (*Figure 8—figure supplement 1C*) and found that the surface expressed GluK1/Neto2 was partially colocalized with presynaptic marker VGLUT1 (*Figure 8—figure supplement 1D*). Together with the above electrophysiological findings that: (1) the aEPSCs from the experimental cells were not significantly reduced by GluK1 specific inhibitor ACET (*Figure 4A–C*); (2) the critical mutants Neto1S3Y/A and Neto2S4T/A impair GluK1 synaptic expression while maintain

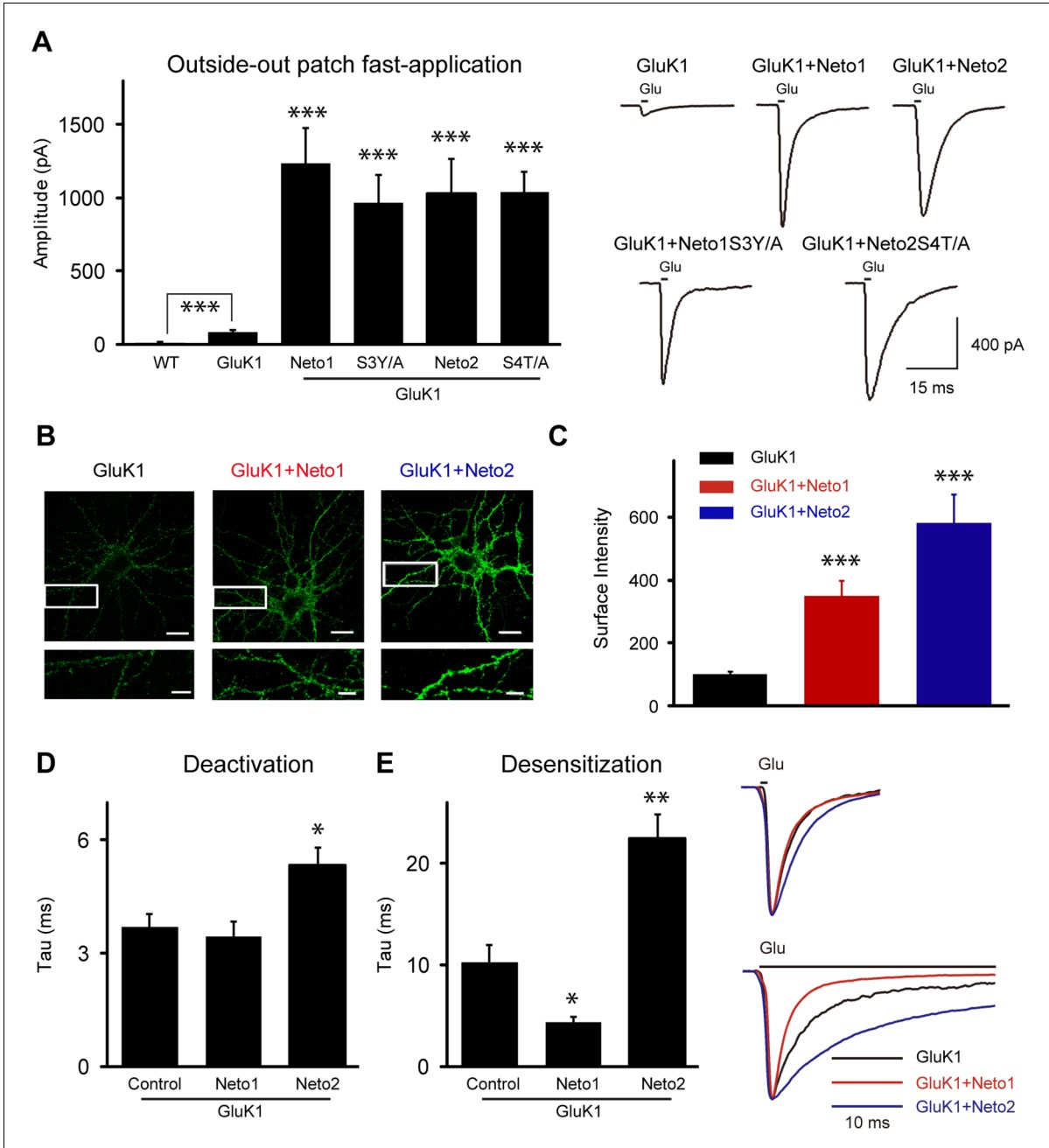

**Figure 8.** Neto1 and Neto2 increase GluK1 receptor surface trafficking and biophysical properties. (**A**) Bar graphs show the amplitude of GluK1 currents (mean ± SEM) from outside-out patches pulled from wild type and transfected CA1 neurons with indicated plasmids and exposed to 1 or 100 ms applications of 10 mM glutamate and 100 μM GYKI53655 (WT, n=7, 8.57 ± 2.51 pA, *** p < 0.0005; GluK1: n=10, 81.65 ± 11.26 pA; GluK1/Neto1: n=9, 1231.94 ± 242.92 pA, *** p < 0.0001; GluK1/Neto1S3Y/A: n=7, 967.14 ± 138.30 pA, *** p < 0.0005; GluK1/Neto2: n=10, 1022.84 ± 241.81 pA, *** p < 0.0005; GluK1/Neto2S4T/A: n=9, 1035.22 ± 115.00 pA, *** p < 0.0001). All the statistical analyses are compared to GluK1 single overexpression using Mann-Whitney U-test. Sample traces and scale bar are shown to the right. (**B**) DIV 10 neurons were transfected with HA-GluK1 and Neto1 or Neto2, as indicated. At DIV 13, cells were stained for surface GluK1 and the intensity of surface GluK1 was quantitated (3 dendrites per neuron) using Metamorph analysis software. Scale bar, 20 μm. Images at the bottom of each panel are higher magnification from the enclosed region. Scale bar, 5 μm. (**C**) Bar graph shows the surface expression of GluK1 (mean ± SEM) from three independent experiments (GluK1: n=34; GluK1/Neto1: n=33; GluK1/Neto2: n=34). An unpaired two-tailed t-test was used to determine the significance of the data: *** p < 0.0001. (**D** and **E**) Bar graphs show mean ± SEM GluK1 deactivation (d, GluK1: n=10, 3.7 ± 0.35 ms; GluK1/Neto1: n=7, 3.43 ± 0.42 ms, p > 0.05; GluK1/Neto2: n=10, 5.33 ± 0.46 ms, * p < 0.05) and desensitization (e, GluK1: n=6, 12.42 ± 2.26 ms; GluK1/Neto1: n=8, 4.93 ± 0.59 ms, * p < 0.05; GluK1/Neto2: n=7, 27.49 ± 3.26 ms, ** p < 0.005) from outside-out patches pulled from indicated transfection CA1 neurons and exposed to 1 or 100 ms applications of 10 mM glutamate and 100 μM

*Figure 8 continued on next page*

*Figure 8 continued*

GYKI53655, respectively. All the statistical analyses are compared to GluK1 single overexpression using Mann-Whitney U-test. Sample traces are shown to the right and are peak-normalized.

The following figure supplement is available for figure 8:

**Figure supplement 1.** GluK1 receptor is localized at synapse.

its surface expression (*Figure 6G*, *7H* and *8A*), it strongly supports the notion that the GluK1 receptors are indeed localized at the synapses and mediated synaptic transmission.

In addition to measuring the size of the peak currents, we also measured the rate of deactivation and desensitization, parameters that might be affected by Neto proteins. Neto1 had no effect on the deactivation of GluK1-mediated currents, but Neto2 did slow deactivation (*Figure 8D*). By contrast, Neto1 enhanced the rate of desensitization, whereas Neto2 slowed the rate of desensitization (*Figure 8E*), in agreement with previous results (*Copits et al., 2011*).

## Discussion

We have selected the CA1 pyramidal cell to study the biology of KARs and their auxiliary Neto subunits. Since the excitatory synapses onto these neurons normally lack KARs, these synapses provide a powerful model to explore the basic mechanisms for expression and targeting of KARs to excitatory synapses. We find that GluK1 by itself is poorly expressed on the neuronal surface and is not present at excitatory synapses. However, in the presence of either Neto1 or Neto2, GluK1 is expressed at very high levels on the neuronal surface and is also present at excitatory synapses. In addition to their role in GluK1 surface trafficking, Neto1 and Neto2 regulate its synaptic targeting independently and the underlying mechanisms for these two processes are different. Interestingly, GluK1 is excluded from synapses expressing AMPARs and is selectively incorporated into silent synapses. Neto2, but not Neto1, slows GluK1 deactivation, whereas Neto1 speeds GluK1 desensitization and Neto2 slows desensitization. These experiments provide important basic information on the mechanisms by which Neto auxiliary subunits control the synaptic incorporation of KARs and their biophysical properties.

### Surface KARs

Expression of GluK1 by itself results in very little KAR surface currents in CA1 pyramidal cells. By contrast co-expression of GluK1 with either Neto1 or Neto2 generates currents approaching a nA in outside out patches. This effect cannot be explained by changes in desensitization, because Neto1 actually increases desensitization and yet generates current of similar magnitude to Neto2, which slows desensitization. In addition, we used ultra fast application to avoid this possibility. An increase in single channel conductance and/or open probability could contribute to the enhanced currents, but are highly unlikely to account for the massive currents recorded with both Neto1 and Neto2. Finally, the surface staining of GluK1 was unequivocally enhanced when Neto proteins are coexpressed. All these results demonstrate the critical role of Netos in the delivery of GluK1 to the surface. Neto2 has no effect on the surface expression of GluK2 in oocytes (*Zhang et al., 2009*). The closest comparison to the present result are those of Copits et al., *Copits et al. (2011)* who found that Neto2, and to a much lesser degree Neto1, enhanced peak GluK1-mediated currents in HEK cells and Neto2, but not Neto1, enhanced surface staining for GluK1 in neurons. This difference between our results and those of Copits et al., *Copits et al. (2011)* might be because different isoforms of GluK1 were used in this and our studies. Sequence comparison of the two isoforms suggests that the intracellular C-tail of GluK1 might be involved in Neto1-regulated surface trafficking. And in agreement with that study (*Copits et al., 2011*), we found that Neto1 enhanced the rate of GluK1 desensitization while Neto2 greatly slowed the rate of desensitization. The mechanism by which Netos modulate delivery is unclear. It could indicate that Neto proteins serve as chaperones and are required for the proper folding and maturation of the KARs, analogous to the role of TARPs in AMPAR trafficking (*Jackson and Nicoll, 2011*). Alternatively or additionally, Neto proteins could play a more direct role in delivering mature receptors to the surface.

## Synaptic KARs

Although CA1 pyramidal cells express functional GluK2 surface receptors (*Bureau et al., 1999*), synaptic KARs are absent from excitatory synapses (*Bureau et al., 1999*; *Castillo et al., 1997*; *Granger et al., 2013*). This absence is not due to the lack of either Neto1 or Neto2 because expression of these proteins did not lead to the appearance of synaptic KARs. Thus either the level of KAR expression is insufficient for synaptic targeting or some other critical protein is missing from CA1 pyramidal cells. Expression of GluK1 also failed to generate synaptic KAR currents, although it was expressed on the surface, albeit at low levels. With either Neto1 or Neto2, GluK1 generated large synaptic currents. This was accompanied with large expression of receptors on the cell surface. There are two possibilities to account for the presence of KARs at synapses. First, the density of the receptors on the surface could be so high that they simply flood the synapse, without any specific targeting signal. Second, their presence at synapses requires a separate targeting mechanism. We believe the latter is the case. We found that critical mutants of Neto1 and Neto2 in their C-terminal domains, which severely limited the synaptic accumulation of GluK1 receptors, had little or no effect on GluK1 surface expression.

Expression of GluK1 and Neto caused a large increase in aEPSC frequency, but no change in aEPSC amplitude. Furthermore, the GluK1 antagonist ACET, blocked the increase in frequency but had no effect on aEPSCs amplitude. And comparing to the wild type Neto proteins, the critical mutants Neto1S3Y/A and Neto2S4T/A promote GluK1 surface trafficking to the same extent but both impair its synaptic expression. Moreover, the Neto2/GluK1 receptors are partially colocalized with presynpatic marker VGLUT1. All these findings suggest that the GluK1 receptors are indeed localized at the synapse and the synaptic GluK1 responses are not due to the spread of glutamate from the synapse. Moreover, these results indicate that KARs and AMPARs do not co-localize at the same synapse, either by adding additional receptors to the synapse or by redistributing synaptic AMPARs. Instead, they appear to selectively populate previously silent synapses, i.e. synapses with NMDARs but no AMPARs. This model raises two sets of intriguing questions. First, what accounts for the fact that on average KAR synapses generate aEPSCs identical in size to AMPAR synapses? This observation suggests a homeostatic process, although the synaptic expression of KARs occurs in the absence of synaptic activity. Second, we know that during LTP an individual synapse, which contains AMPARs before LTP, can accumulate additional AMPARs during LTP (*Harvey and Svoboda, 2007*; *Lee et al., 2009*; *Matsuzaki et al., 2004*; *Oliet et al., 1996*). We also know that expressed GluK1 receptors at CA1 synapses on an AMPAR null background exhibit normal levels of LTP (*Granger et al., 2013*). These findings raise a number of interesting questions. Does the genetic removal of AMPARs from the synapse now allow the synapse to accept KARs? Although KARs are excluded from synapses that express AMPARs, can LTP drive KARs into AMPAR containing synapses?

Previous studies have reported that Neto1 is involved in synaptic NMDAR function although the findings were inconsistent. In one study it was found that Neto1 is critical for NMDAR subunit NR2A synaptic expression in CA1 neuron (*Ng et al., 2009*), but another study showed that NR2B but not NR2A synaptic expression is increased in CA3 neurons of Neto1 knock-out mice (*Wyeth et al., 2014*). However, here we found neither Neto1 nor Neto2 itself has any effect on NMDAR EPSC. Both GluK1/Neto1 and GluK1/Neto2 coexpression increased the size of the NMDAR EPSC, although this increase was much more modest than the EPSC observed at −70 mV. This could result from a modest synaptogenic effect. Although we did not observe an increase in spine density, the modest effects might be difficult to see with our imaging. Alternatively synapses could be added to the shaft and thus not visible in our spine density quantification.

## Neto domain structure required for synaptic trafficking

We sought to define the critical domain(s) of Neto1 and Neto2 required for the synaptic trafficking of GluK1. For Neto1 the critical region is the last 20 amino acids. Except for the PDZ ligand domain, which is not required, there is no obvious homology to known protein-protein interaction domains. There are putative phosphorylation sites in this region and their mutation disrupts trafficking. For Neto2 the critical region was located in the middle of the C-terminal domain and could be narrowed down to a 12 amino acid stretch. Again there is no obvious homology of this region to known protein-protein binding motifs. There are putative phosphorylation sites in this region, which when

mutated disrupt synaptic trafficking of KARs. It will be of interest in future studies to determine the potential roles of phosphorylation of Neto1 and Neto2 and the involved kinase(s) in GluK1 synaptic trafficking.

In summary, this study has characterized the properties of Neto1 and Neto2 in controlling GluK1 receptor synaptic trafficking in hippocampal neurons. We have selected an excitatory synapse that normally does not express KARs, in order to determine the minimal requirements that govern the insertion of KARs into excitatory synapses. Our results demonstrate that Neto auxiliary proteins have two functionally distinct roles in the biology of the GluK1 type of KAR: First, they are essential for the delivery of receptors to the surface and for their targeting to the synapse. Second, they modify the gating kinetics of GluK1. These properties are reminiscent of those of TARPs, which perform remarkably similar roles in the biology of AMPARs. It will be interesting to see how many of the properties we describe at CA1 synapses are held in common with excitatory synapses that normally express KARs, e.g. hippocampal mossy fiber synapses.

# Materials and methods

## Experimental constructs

The cDNAs of rat GluK1 (gift from Dr. Stephen F. Heinemann), mouse Neto1 (purchased from Open Biosystems) and rat Neto2 (gift from Dr. Susumu Tomita) as well as the Neto1 and Neto2 mutants were subcloned into pCAGGS vector for biolistic transfection.

## Electrophysiology in slice cultures

Organotypic hippocampal slice cultures were made as previously described (*Schnell et al., 2002*). Slices from P6-P8 rats were biolistically transfected with indicated plasmids together with FUGW-EGFP plasmid as a tracer on DIV 2 and then on DIV 8 dual whole-cell recordings in area CA1 were done by simultaneously recording responses from a fluorescent transfected neuron and neighboring untransfected control neuron. Pyramidal neurons were identified by morphology and location. Series resistance was monitored on-line, and recordings in which series increased to >30 MOhm or varied by >50% between neurons were discarded. Dual whole-cell recordings measuring evoked EPSCs used artificial cerebrospinal fluid (ACSF) bubbled with 95% $O_2$/5% $CO_2$ consisting of (in mM) 119 NaCl, 2.5 KCl, 4 $CaCl_2$, 4 $MgSO_4$, 1 $NaH_2PO_4$, 26.2 $NaHCO_3$, 11 Glucose. 100 µM picrotoxin was added to block inhibitory currents and 4 µM 2-Chloroadenosine was used to control epileptiform activity. Intracellular solution contained (in mM) 135 $CsMeSO_3$, 8 NaCl, 10 HEPES, 0.3 EGTA, 5 QX314-Cl, 4 MgATP, 0.3 $Na_3GTP$, 0.1 spermine. A bipolar stimulation electrode was placed in stratum radiatum, and responses were evoked at 0.2 Hz. Peak AMPAR and GluK1 currents were recorded at −70 mV, and NMDAR current amplitudes 100 ms following the stimulus were recorded at +40 mV. Paired-pulse ratio was determined by delivering two stimuli 40 ms apart and dividing the peak response to stimulus 2 by the peak response to stimulus 1. All these data were analyzed off-line with custom software (IGOR Pro, free download from following site: https://www.wavemetrics.com/order/order_igordownloads.htm). For $Sr^{2+}$-evoked asynchronous EPSC recording, the ACSF was the same as above with the equimolar substitution of $SrCl_2$ for $CaCl_2$. 100 µM picrotoxin was also included but without 2-Chloroadenosine. Stimulation was increased from 0.2 Hz to 2 Hz to optimize the frequency of $Sr^{2+}$-evoked responses (*Oliet et al., 1996*). $Sr^{2+}$-evoked aEPSCs were analyzed off-line with custom IGOR PRO software, and in all cases at least 100 quantal events were used. For fast application, somatic out-side out patches were excised from wild type or transfected CA1 pyramidal neurons using 3–5 MΩ pipettes. The fast responses to glutamate were recorded at −70 mV. Glutamate pulses of 1 or 100 ms were applied to patches by a theta-glass pipette every 10–20 s using a piezoelectric controller (Siskiyou) (*Shi et al., 2009*). Glutamate (10 mM) was dissolved in the HEPES ACSF consisting of (in mM) NaCl 140, KCl 5, $MgCl_2$ 1.4, $CaCl_2$ 1, EGTA 5, HEPES 10, $NaH_2PO_4$ 1, D-glucose 10, with pH adjusted to 7.4, with the addition of 100 µM D-APV, 0.5 M tetrodotoxin and 100 µM GYKI53655 to isolate GluK-mediated currents. The control barrel contained the same HEPES ACSF with all the inhibitors and 1 mM sucrose but except glutamate. The open-tip response had a switch on and off time of less than 200 µs. Responses were collected with a Multiclamp 700A amplifier (Axon Instruments), filtered at 2 kHz, and digitized at 10 kHz.

## Anatomy and imaging

Slice cultures were maintained and transfected as described above and on DIV 8 a transfected CA1 pyramidal neuron and a wild type one were patched simultaneously and filled with Alexa Fluor 568 dyes through the patch pipette for about 15–20 min. After filling, slices were fixed in 4% PFA/4% sucrose in PBS for 30 min at room temperature, followed by washing at least three times with PBS. Then slices were mounted and imaged by using super-resolution microscopy (N-SIM Microscope System, Nikon). The experimental cells were identified by GFP fluoresces. Images along the stretch of CA1 pyramidal neuron primary apical dendrite from 100 μm to 200 μm from the cell body were acquired with a 100x oil objective in 3D-SIM mode using supplied SIM grating (3D EX V-R 100x/1.49) and processed and reconstructed using supplied software (NIS-Elements, Nikon). Spine density analysis was performed manually on individual sections using ImageJ.

## Surface immunolabeling/imaging

For determining surface expression, an N-terminal HA tag was inserted after the signal peptide in GluK1. DIV 10 rat hippocampal cultures were transfected with HA-GluK1 alone or together with Neto1 or Neto2. At DIV 13, surface GluK1 (green) was labeled with a rabbit HA antibody (Abcam, Cat. No. ab9110) at room temperature for 10 min, followed by Alexa-488 secondary antibody (Life technologies, A11034). The images were captured as Z-stacks using a 63X oil immersion objective of LSM 510 Meta Zeiss confocal microscope. A projection image was created using different optical sections (0.35 μm) and is presented. To determine changes in surface expression, the amount of surface GluK1 divided by the area of the ROI was calculated from 3 dendritic regions per neuron using Metamorph. The data presented is mean ± SEM from three independent experiments.

## Statistical analysis

Significance of evoked dual whole-cell recordings and aEPSC frequency compared to controls was determined using the two-tailed Wilcoxon signed-rank sum test. For all experiments involving un-paired data, including all outside-out patch data, a Mann-Whitney U-test with Bonferonni correction for multiple comparisons was used. Paired-pulse ratios and spine densities were analyzed with unpaired t test. Data analysis was carried out in Igor Pro (Wavemetrics), Excel (Microsoft), and GraphPad Prism (GraphPad Software).

# Acknowledgements

This work was funded by grants from the US NIMH (R.A.N.) and both KWR and RML were supported by the Intramural Research Program of NINDS. YSS is supported by Grants from Natural Science Foundation of China (31371061 and 31571060) and the Ministry of Science and Technology of China (2014CB942804). We are grateful to K Bjorgan, M Cerpas and D Qin for technical assistance, and all members of the Nicoll laboratory for discussion of and comments on the manuscript.

# Additional information

### Funding

| Funder | Grant reference number | Author |
|---|---|---|
| National Institute of Mental Health | 5R01MH080379-09, 5R37MH038256-32 and 2R01MH070957-11A1 | Roger A Nicoll |
| National Institute of Neurological Disorders and Stroke | NS002994-13 | Katherine W Roche |
| National Natural Science Foundation of China | 31371061 | Yun S Shi |
| Ministry of Science and Technology of the People's Republic of China | 2014CB942804 | Yun S Shi |

| National Natural Science Foundation of China | 31571060 | Yun S Shi |
|---|---|---|

The funders had no role in study design, data collection and interpretation, or the decision to submit the work for publication.

## Author contributions

NS, designed experiments, performed all electrophysiology and all imaging in slice, constructed all new constructs, and conducted data analysis; wrote the manuscript, Conception and design, Acquisition of data, Analysis and interpretation of data, Drafting or revising the article, Contributed unpublished essential data or reagents; YSS, Initiated the project and generated preliminary data; provided manuscript comments and directed revisions, Conception and design, Acquisition of data, Analysis and interpretation of data; RML, Designed and carried out imaging in dissociated neurons, and conducted data analysis; provided manuscript comments and directed revisions, Acquisition of data, Analysis and interpretation of data, Contributed unpublished essential data or reagents; KWR, Helped design experiments, interpret data, and supervised the project; provided manuscript comments and directed revisions, Analysis and interpretation of data, Drafting or revising the article, Contributed unpublished essential data or reagents; RAN, Helped design experiments, interpret data, and supervised the project; wrote the manuscript, Conception and design, Analysis and interpretation of data, Drafting or revising the article, Contributed unpublished essential data or reagents

## Author ORCIDs

Nengyin Sheng, http://orcid.org/0000-0003-3086-3950

## Ethics

Animal experimentation: This study was performed in strict accordance with the recommendations in the Guide for the Care and Use of Laboratory Animals of the National Institutes of Health. All of the animals were housed in rodent housing at the UCSF and the animal health was monitored by a staff that includes the attending veterinarian, a veterinary pathologist, and a team of veterinary nurses. The University of California, San Francisco has on file with the Office of Protection from Research Risks, National Institutes of Health, U.S. Public Health Service (PHS), an approved Assurance of Compliance with PHS Policy on Humane Care and Use of Laboratory Animals by Awardee Institutions (#3400-01). That document expresses UCSF's commitment to comply with PHS policy and all applicable laws and regulations regarding the care and use of laboratory animals in research and instruction.

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
