## [Decision Letter]

Thank you for submitting your work entitled "Neto auxiliary proteins control both the trafficking and biophysical properties of the kainate receptor GluK1" for consideration by *eLife*. Your article has been favourable evaluated by Gary Westbrook (Senior Editor) and two reviewers, one of whom is a member of our Board of Reviewing Editors.

The reviewers have discussed the reviews with one another and the Reviewing editor has drafted this decision to help you prepare a revised submission.

Summary:

Sheng et al. provides information on the role of the auxiliary subunits Neto 1 and 2 on the kinetic properties and surface trafficking of kainate (KA) receptors in CA1 principal cells. These cells are ideally suited for this investigation because they do not endogenously express GluK1 receptor subunits. The authors show that over-expression of Neto1 and Neto2 together with GluK1 increases EPSC amplitude. Interestingly, K1 receptors are selectively introduced into silent synapses, but excluded from synapses containing AMPA receptors. Both, Neto 1 and 2 show distinct effects on the deactivation and desensitization kinetics of KA receptors. Thus, the two auxiliary subunits Neto1 and 2 control KA and synaptic properties.

Essential revisions:

Your manuscript was carefully reviewed by two experts. Both value your study on the role of the auxiliary subunits Neto1 and 2 on the kinetic properties of KA receptors and their surface trafficking as a internally very consistent study. We think that addressing the below localization and kinetic issues would enhance the value and interpretation of the experiments.

1) Reviewer 1 praised the idea to study the effects of Neto on KA trafficking in CA1 principal cells, which usually do not endogenously express GluK1 receptor subunits. However, we would like to see more precise data on localization of KA1. The images in Figure 8 showing the enhanced intensity in Ab labeling are fine, but a better demonstration, e.g. using electron microscopy, would better provide better evidence of extrasynaptic versus synaptic localization of KA1 as well as the increased surface expression at postsynaptic sites. Alternatively, freeze fracture analysis could provide quantitative data on the number and distribution of KA1 in the presence and absence of Neto1 or 2.

2) Reviewer's 2 main question was whether KA receptors need to be "synaptic" in order to respond to released glutamate. The glutamate affinity of GluK1 subunits (and likely receptors with Netos and/or GluK5) is ca. 1000x higher than AMPA receptors. Therefore, AMPA and KA receptors may not have to be in the same place to respond equally well to released glutamate. But how close are they? Do the exogenous KA receptors need to reach synapses at all to respond to glutamate? One might expect that KA receptors should respond to much lower glutamate concentrations than AMPA receptors. This question could be addressed with simulations, fast vs slow antagonists or using information about activation kinetics to understand what glutamate concentration is needed to produce the KARs that produce quantal components of the aEPSCs.

---

## [Author Response]

*Essential revisions:*

*Your manuscript was carefully reviewed by two experts. Both value your study on the role of the auxiliary subunits Neto1 and 2 on the kinetic properties of KA receptors and their surface trafficking as a internally very consistent study. We think that addressing the below localization and kinetic issues would enhance the value and interpretation of the experiments. 1) Reviewer 1 praised the idea to study the effects of Neto on KA trafficking in CA1 principal cells, which usually do not endogenously express GluK1 receptor subunits. However, we would like to see more precise data on localization of KA1. The images in Figure 8 showing the enhanced intensity in Ab labeling are fine, but a better demonstration, e.g. using electron microscopy, would better provide better evidence of extrasynaptic versus synaptic localization of KA1 as well as the increased surface expression at postsynaptic sites. Alternatively, freeze fracture analysis could provide quantitative data on the number and distribution of KA1 in the presence and absence of Neto1 or 2.*

We appreciate the reviewer’s interest in our study and his/her suggestions for improvements. He/she focuses on a frustrating aspect in the study of KARs. The lack of good antibodies has plagued the field since the beginning. We have scoured the field and are unable to identify any good antibody for the wild type GluK1 subunit. This has lead researchers to tag the receptors. We have used an N-terminal HAtagged GluK1 construct to label our cells, and demonstrate that Neto proteins greatly enhance the surface expression of GluK1. However, this construct is of limited value in immunocytochemical localization studies. We overexpressed this HA-GluK1 receptor together with Neto1 and then carried out electrophysiological examination of its surface expression by puffing L-glutamate or kainate and recording the whole cell currents. GYKI153655 (100 μM), APV (100 μM), pictrotoxin (100 μM) and TTX (0.5 μM) were present during these experiments. Our physiological results (see Figure 8—figure supplement 1) indicate that the construct does get to the surface as expected from the immunofluorescence microscopy results. However, much to our surprise, this HAtagged receptor is poorly, if at all, targeted to the synapse (see Figure 8—figure supplement 1). Similar results were obtained with another previously reported Myc-tagged GluK1 receptor (see Figure 8—figure supplement 1). This is remarkably similar to our unpublished results with the GFP tagging of the GluA1 receptor subunit, i.e., the receptor is highly expressed on the cell surface but is excluded from the synapse. The results discussed above as well as numerous additional experiments indicate a crucial role of the NTD in the synaptic targeting of receptors.

This is a long way of saying that there are no adequate anatomical tools available for studying the subcellular localization of our expressed GluK1. We did try a less direct approach to label synaptic GluK1. Putting the Myc-tag at the Neto2 N-terminus after the signal peptide does not interfere with its ability to traffic GluK1. Since Neto2 is required for synaptic localization of GluK1, we co-expressed wild type GluK1 with a tagged Neto2. The assumption is that if the tagged Neto2 can drive the GluK1 to the synapse, which it does (Figure 8—figure supplement 1), it would imply that GluK1, as well as the tagged Neto2, is present at the synapse. We tried this approach first with slice cultures. Unfortunately, we were unable to get robust enough signals in this experiment to feel comfortable with this approach. Then we expressed the Myc-Neto2 by itself or co-expressed it with GluK1 in dissociated hippocampal neurons. We found that the expression of Myc-Neto2 by itself is relatively low (see Figure 8—figure supplement 1), but when coexpressed with GluK1, the surface expression of Myc-Neto2 is greatly increased (see Figure 8—figure supplement 1), suggesting that the GluK1 is necessary for the stabilization of its auxiliary Neto proteins. We also co-immunostained with an antibody against the presynaptic marker VGLUT1 and found that it partially co-localized with surface Myc-Neto2 (see Figure 8—figure supplement 1). We provide these images for the reviewer, but would be happy to provide them as supplemental figures. We should also point out that these experiments could not be done with tagged Neto1, because this construct did not express.

2) Reviewer's 2 main question was whether KA receptors need to be "synaptic" in order to respond to released glutamate. The glutamate affinity of GluK1 subunits (and likely receptors with Netos and/or GluK5) is ca. 1000x higher than AMPA receptors. Therefore, AMPA and KA receptors may not have to be in the same place to respond equally well to released glutamate. But how close are they? Do the exogenous KA receptors need to reach synapses at all to respond to glutamate? One might expect that KA receptors should respond to much lower glutamate concentrations than AMPA receptors. This question could be addressed with simulations, fast vs slow antagonists or using information about activation kinetics to understand what glutamate concentration is needed to produce the KARs that produce quantal components of the aEPSCs.

These are good points, which were not addressed adequately in the original manuscript. The reviewer brings up the issue of “synaptic”, which is viewed somewhat differently by physiologists and anatomists. For anatomists the answer is clear, it’s the PSD. For physiologists it is a bit less clear. For instance, mGluRs are not present in the PSD, but form an annulus around it. Yet in the cerebellum most people would agree that parallel fiber activation generates a slow EPSP. However, the point that the reviewer raises is how far from the release site can GluK1 receptors be and still be activated by synaptically released glutamate. The strongest argument that our synaptic GluK1 responses are not due to the spread of glutamate from the synapse is the following. If glutamate were able to activate extrasynaptic receptors, then one would expect that the amplitude of both eEPSCs and aEPSCs to be reduced by the GluK1 antagonist ACET and yet there is no change. The second piece of evidence is that the Neto1S3Y/A and Neto2S4T/A mutants both promote expression of GluK1 on the surface to the same extent as the wild type Neto1 and Neto2 and yet there are severe defects in synaptic responses, strongly suggesting that, while the receptor is abundant at extrasynaptic sites, synaptically released glutamate does not have access to these receptors. We have added a section in the Discussion addressing these important issues and thank the reviewer for alerting us to these issues.